# Study of Information Dissemination in Hypernetworks with Adjustable Clustering Coefficient

Pengyue Li [1,2,3], Liang Wei [2,3,4], Haiping Ding [1,2,3], Faxu Li [1,2,3,*] and Feng Hu [1,2,3]

1 School of Computer, Qinghai Normal University, Xining 810008, China
2 The State Key Laboratory of Tibetan Intelligent Information Processing and Application, School of Computer, Xining 810008, China
3 Academy of Plateau Science and Sustainability, Qinghai Normal University, Xining 810016, China
4 School of Mathematics and Statistics, Qinghai Normal University, Xining 810008, China
* Correspondence: lifaxu@qhnu.edu.cn; Tel.: +86-139-9707-7193

**Abstract:** The structure of a model has an important impact on information dissemination. Many information models of hypernetworks have been proposed in recent years, in which nodes and hyperedges represent the individuals and the relationships between the individuals, respectively. However, these models select old nodes based on preference attachment and ignore the effect of aggregation. In real life, friends of friends are more likely to form friendships with each other, and a social network should be a hypernetwork with an aggregation phenomenon. Therefore, a social hypernetwork evolution model with adjustable clustering coefficients is proposed. Subsequently, we use the SIS (susceptible–infectious–susceptible) model to describe the information propagation process in the aggregation-phenomenon hypernetwork. In addition, we establish the relationship between the density of informed nodes and the structural parameters of the hypernetwork in a steady state using the mean field theory. Notably, modifications to the clustering coefficients do not impact the hyperdegree distribution; however, an increase in the clustering coefficients results in a reduced speed of information dissemination. It is further observed that the model can degenerate to a BA (Barabási–Albert) hypernetwork by setting the clustering coefficient to zero. Thus, the aggregation-phenomenon hypernetwork is an extension of the BA hypernetwork with stronger applicability.

**Keywords:** hypernetwork; social networks; information dissemination; clustering coefficient; SIS model

## 1. Introduction

The dissemination of information has greatly contributed to the advancement of human society. As science and technology continue to evolve, the integration of the Internet into people's daily life leads to an increasing frequency of information exchange. In social networks [1–4], users' interdependence and cooperation serve as the basis for facilitating information dissemination. Therefore, identifying the mechanisms and characteristics of information diffusion is essential for both theoretical and practical purposes.

Information dissemination refers to the process of spreading information from a source node to other nodes within a network, similar to how infectious diseases spread in populations. Research on propagation has focused on the study of network models and the study of propagation mechanisms. Network models include dynamic networks [5–7], scale-free networks [8,9], node-weighted networks [10], edge-weighted networks [11], variable-growth networks [12], and correlation networks [13]. The study of propagation mechanisms includes SIS [14], SIR [15], etc. In cooperative complex networks, nodes represent individuals and edges represent cooperative relationships between individuals, such as scientific cooperative networks [16], corporate cooperative networks [17], and actor cooperative networks [18].

Previous studies have made significant contributions to the advancement of the field of spreading dynamics, and information propagation models based on complex networks

have been extensively explored. However, the complexity of real-life systems is often higher than what can be represented by the simple pairwise relations present in complex networks, calling for more intricate relationship descriptions. As a result, complex networks may not be able to fully capture the group characteristics in social networks.

An alternative model for information propagation is the hypernetwork [19], which is constructed using the hypergraph theory. Unlike edges in conventional networks, hyperedges in hypernetworks can accommodate a wide range of nodes and depict higher-order interactions among them. Estrada and Rodríguez-Velázquez [20] were the pioneers in the development of hypernetworks. They first introduced the concept and extended the notions of clustering coefficient and subgraph centrality to the hypernetwork model. Since then, several extensions to the model have been proposed, including directed hypernetworks by Antonio P. Volpentesta [21], hyperstructures by Criado [22], and node importance metrics by Xiao [23]. Furthermore, the topological characteristics of hypernetworks have been thoroughly investigated by Ma and Liu [24]. In terms of dynamic evolution models, Guo [25] proposed a nonuniform evolving hypergraph model with a nonlinear preferential attachment and an attractiveness. Wang [26] proposed a hypernetwork evolution model that generated new hyperedges through the combination of an old node and several new nodes, while Hu [27] presented a dynamic hypernetwork model that relied on growth and preferential attachment mechanisms. Guo [28] later unified these models to propose a novel hypernetwork evolutionary model where old and new nodes generate multiple new hyperedges. Lastly, Shen [12] proposed variable-growth hypernetworks, which saw the hyperedges and nodes undergo a growth over time.

The application of hypernetworks has led to significant progress in the field of information propagation. However, the earlier works on hypernetwork models reviewed above commonly extend the mechanisms of complex networks for evolution. In actual social networks, clustering usually occurs. Friends of friends are more likely to form friendships, and academics or research teams in the same field are more likely to collaborate, which is not adequately captured by existing hypernetwork models. Thus, Shen [29] proposed a hypernetwork model based on the aggregation phenomenon. However, this model only considered the selection of one old node, while in reality, multiple old nodes and their neighbors may be selected. Therefore, it holds significant importance to delve into and explore the regulations of information dissemination based on this network model. In this paper, we study the hyperdegree distribution and analyzed the relationship with the clustering coefficient and focus on the structural parameters of the hypernetwork, including the number of nodes, clustering coefficient, new nodes, old nodes, new hyperedges, as well as the propagation parameters such as spreading rate, recovering rate, and spreading seed, which all impact information propagation. We compare our model with the BA hypernetwork and find that fundamentally, the BA hypernetwork is an aggregation-phenomenon hypernetwork with a clustering coefficient of zero.

The structure of the paper is as follows. Section 2 introduces the concept of hypernetworks, the evolution process of the agglomeration-phenomenon hypernetwork, and the concepts of the SIS model. In Section 3, the information dissemination model is introduced, and the hyperdegree distribution derivation and theoretical analysis are performed. The last two sections of the paper elaborate on the simulation results and present our conclusions.

## 2. Theoretical Background

### 2.1. The Concept of Hypernetwork

Berge [30,31] proposed the concept of hypergraphs in 1970. The definition of a hypergraph is given below. Let $V = \{v_1, v_2, \ldots, v_n\}$ be a finite set. If $E_i \neq \varnothing$ and $\bigcup_{i=1}^{e} E_i = V$, the pair $H = (V, E)$ is called a hypergraph. Here, the elements $v_1, v_2, \ldots, v_n$ of $V$ are called the nodes or vertices of the hypergraph. $E = \{E_1, E_2, \ldots, E_e\}$ is the set of edges of the hypergraph. The elements of set $E_i$ of set $E$ ($i = 1, 2, \ldots, e$) are called the hyperedges of the hypergraph, and $|E_i|$ denotes the cardinality of the set $E_i$. Moreover, $H = (V, E)$ is a $k$-uniform hypergraph if $|E_i| = k$.

The definition of a hypernetwork [28] is as follows. Suppose that $\Omega = \{(V, E^h) | (V, E^h)\}$ is a finite hypergraph and $G$ is a map from $T = [0, +\infty)$ into $\Omega$. Then, for a given $t \geq 0$, $G(t) = (V(t), E^h(t))$ is a finite hypergraph, where t denotes the time step. A collection of hypergraphs is a hypernetwork $\{G(t), t \in T\}$. The number of hyperedges of the contained node $v_i$ is the hyperdegree of $v_i$. The hyperdegree distribution is the probability distribution of the hyperdegree of all nodes in the hypernetwork.

### 2.2. The Evolution of Agglomeration-Phenomenon Hypernetwork

In a hypernetwork, when two or more nodes are present in the same hyperedge, it can be suggested that they are adjacent to each other. The presence of a hyperedge between neighbors of a node is called the clustering phenomenon. Social hypernetworks often exhibit the phenomenon of agglomeration. Based on the characteristics of such hypernetworks, we propose a hypernetwork model that considers the aggregation phenomenon. The construction steps of the model are as follows:

(i)    Initialization: a hyperedge in the hypernetwork contains $m_0$ nodes.
(ii)   At each time step $t$, $m_1$ new nodes form a new hyperedge with $m_2$ existing nodes, and $m$ nonrepeating hyperedges are constructed at each time step.
(iii)  Hyperdegree preference attachment: The probability $\prod(k_{iu}^h(t))$ that the $m_1$ new nodes connect to the $u_{th}$ old node of the $i_{th}$ batch is proportional to a function of the hyperdegree $k_{iu}^h(t)$, such that:

$$\prod(k_{iu}^h(t)) = \frac{k_{iu}^h(t)}{\sum_w k_w^h(t)} \tag{1}$$

The hyperdegree of the $u_{th}$ node in the $i_{th}$ batch at time $t$ can be represented by $k_{iu}^h(t)$, while the total number of hyperdegrees of all nodes at time $t$ can be represented by $\sum_w k_w^h(t)$. This process is repeated $m_2$ times to select $m_2$ old nodes.
(iv)   Clustering attachment with probability $p$: If two old nodes $u, v$ are selected in the previous preferential attachment step to form a new hyperedge with new nodes, the new nodes randomly select $m_2$ neighbors of $u$ or $v$ to form a new hyperedge. If all neighbors of $u$ or $v$ have been selected and no new nonrepeating hyperedge can be formed, then the preferential attachment step is executed.

As shown in the schematic diagram in Figure 1, the hypernetwork model based on the agglomeration phenomenon is constructed.

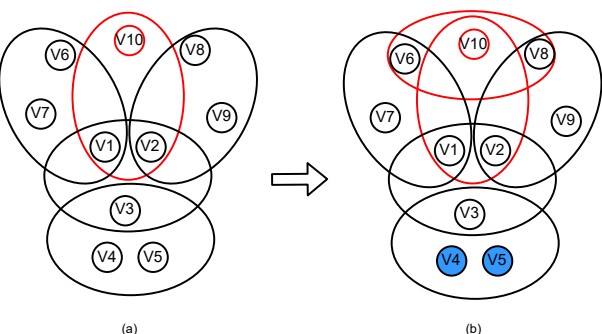

(a)                                        (b)

**Figure 1.** Construction of the aggregation-phenomenon model. New nodes and hyperedges are represented by a red color. At time step $t$, the selection preference attachment is performed as follows: (**a**) New node $v_{10}$ ($m_1 = 1$) preferentially select the existing nodes $v_1$ and $v_2$ ($m_2 = 2$) to form a new hyperedge. (**b**) The new node randomly chooses neighbors $v_6$ and $v_8$ of $v_1$ or $v_2$ to form the second hyperedge with parameter ($m = 2$). Blue nodes $v_4$ and $v_5$ in the figure indicate that they are not available for selection by the new nodes since they are not neighbors of $v_1$ or $v_2$.

### 2.3. SIS Model

The process of information transmission is analogous to that of infectious diseases, and the SIS model may be utilized to describe the propagation and evolution of information.

In this model, individuals are categorized as susceptible (S-state) nodes or informed (I-state) nodes. Whenever an S-state node contacts an I-state node, the information will be transmitted to the I-state node with a probability of $\beta$, and subsequently, the I-state node may revert to the S-state node with a probability of $\gamma$ due to factors such as forgetting. Additionally, a node restored to the S-state may still acquire information and become an I-state node once again.

## 3. Information Dissemination Model Based on RP Strategy

### 3.1. Model Description

The agglomeration phenomenon is common in social systems and friendship networks, and scientific cooperation hypernetworks are no exception. Nodes representing individuals and hyperedges represent social groups formed by those individuals. In the reactive process (RP) strategy, a node is initially selected as an I-state node, which can transmit information to S-state nodes in all its neighbors, and its neighbors obtain information with a certain probability. The dynamic transmission process of information is as follows:

1. Initialization: at $t = 0$, a node is selected as the I-state node, while the remaining nodes are classified as S-state nodes.
2. Propagation: at each subsequent time step, the I-state node transmits information to all neighbor nodes in the S-state. S-state nodes change to I-state nodes with a probability of $\beta$. Conversely, I-state nodes may transition back to the S-state node with a probability of $\gamma$ due to factors such as forgetfulness.
3. Steady state: with the continuous spread of information, the density of nodes in the I-state gradually stabilizes and fluctuates slightly around this stable density, signifying that information propagation in the network has reached a steady state.

The description of the information transmission process is shown in Figure 2.

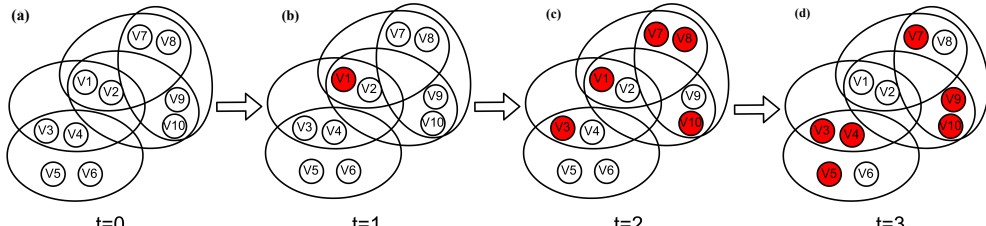

t=0      t=1      t=2      t=3

**Figure 2.** Description of the global information transmission process. Red nodes represent I-state individuals and hollow nodes represent S-state individuals. (**a**) At the initial moment, all nodes are in the S-state. (**b**) When $t = 1$, the $v_1$ node is set as the spreading seed (I-state). (**c**) When $t = 2$, the neighbor nodes $v_3$, $v_7$, $v_8$, and $v_{10}$ of $v_1$ are notified to change to the I-state. (**d**) When $t = 3$, nodes $v_4$, $v_5$, and $v_9$ transition to the I-state, whereas nodes $v_1$ and $v_8$ revert to the S-state.

### 3.2. Theoretical Analysis

By employing techniques of the mean field theory, the following dynamical equation can be derived for $k_{ij}^h(t)$ at the preferential attachment step

$$\frac{\partial_t k_{ij}^h}{\partial_t} = \lambda[m - (m-1)p]m_2 \frac{k_{ij}^h}{\sum_w k_w^h} \qquad (2)$$

where $\lambda$ is the rate of node batches' arrival. If $(m-1)p$ is the number of hyperedges generated by the clustering attachment, then $m - (m-1)p$ is the number of hyperedges generated by the preferential attachment, and the above equation multiplied by $m_2$ is the total number of old nodes selected by the preferential attachment.

When a new node connects with an old node $u$ and the old node is also a neighbor of another node $j$, the second node $j$ is selected at random to form a new hyperedge with the

new node. During this process, $k_{ij}^h(t)$ is governed by the following dynamical equation at the aggregation attachment step:

$$\frac{\partial_t k_{ij}^h}{\partial_t} = \lambda(m-1)pm_2 \sum_{u \in j} \Omega_j \frac{k_{iu}^h}{\sum_w k_w^h} \frac{1}{(m_1-1)k_{iu}^h} = \lambda(m-1)pm_2 \frac{k_{ij}^h}{\sum_w k_w^h} \tag{3}$$

where $\Omega_j$ represents the set of neighbors of node $j$. Similar to the idea discussed in Equation (2), $(m-1)pm_2$ is the total number of old nodes selected for the clustering attachment.

Utilizing Equations (2) and (3), the dynamical equation for the total rate of change in the hyperdegree of node $j$ within a single time step can be derived:

$$\frac{\partial_t k_{ij}^h}{\partial_t} = \lambda[m-(m-1)p]m_2 \frac{k_{ij}^h}{\sum_w k_w^h} + \lambda(m-1)pm_2 \frac{k_{ij}^h}{\sum_w k_w^h} = \lambda m m_2 \frac{k_{ij}^h}{\sum_w k_w^h} \tag{4}$$

If $m_1$ new nodes and $m_2$ existing nodes form a new hyperedge, the hyperdegree of the respective node increases by $(m_1+m_2)$ at each time step. At moment $t$, a total of $m$ new hyperedges are formed. Using the notation $\sum_{ij} k_{ij}^h(t) \approx m(m_1+m_2)E[N(t)] = \lambda m(m_1+m_2)t$, it follows that

$$\frac{\partial_t k_{ij}^h}{\partial_t} = \lambda m m_2 \frac{k_{ij}^h}{m(m_1+m_2)t} = \frac{m_2 k_{ij}^h}{(m_1+m_2)t} \tag{5}$$

The initial condition for Equation (5) conforms to the following equation:

$$k_{ij}^h(t_i) = m \tag{6}$$

where $t_i$ represent the time at which the $i_{th}$ batch node enters the hypernetwork. Furthermore, the node of the $i_{th}$ batch emerges at time $t_i$. Therefore, $k_{ij}^h(t_i)$ denotes the hyperdegree of the $j_{th}$ node in the $i_{th}$ batch at the moment of emergence.

Using Equation (6), we can obtain the solution to Equation (5) as follows

$$k_{ij}^h(t_i) = m \left(\frac{t}{t_i}\right)^{\frac{m_2}{m_1+m_2}} \tag{7}$$

Through Equation (7), we obtain

$$P(k_{ij}^h(t) \geq k) = P(m((\frac{t}{t_i})^{\frac{m_2}{m_1+m_2}}) \geq k) = P(t_i \leq (\frac{m}{k})^{\frac{m_1}{m_2}+1}t) \tag{8}$$

It is worth observing that the process of node batch arrivals follows a Poisson process with rate $\lambda$. Thus, the time $t_i$ follows a gamma distribution with parameters $(i, \lambda)$, which leads to the following expression

$$P(k_{ij}^h(t) \geq k) = P(m((\frac{t}{t_i})^{\frac{m_2}{m_1+m_2}}) \geq k) = P(t_i \leq (\frac{m}{k})^{\frac{m_1}{m_2}+1}t) \tag{9}$$

By using Equations (8) and (9), we have

$$P(t_i \leq (\frac{m}{k})^{\frac{m_1}{m_2}+1}t) = 1 - e^{-(\frac{m}{k})^{\frac{m_1}{m_2}+1}\lambda t} \sum_{l=0}^{i-1} \frac{[(\frac{m}{k})^{\frac{m_1}{m_2}+1}\lambda t]^l}{l!} \tag{10}$$

By substituting Equation (10) into the expression, the transient hyperdegree distribution of this hypernetwork can be derived as follows

$$P\{k_{ij}^h, t\} \approx \frac{1}{m_1 E[N(t)]} \sum_{ij} P(k_{ij}^h(t) = k)$$

$$= (\frac{m_1}{m_2} + 1)(\frac{m}{k})^{\frac{m_1}{m_2}+2} \frac{\lambda t}{m} \frac{(\lambda t(m/k)^{\frac{m_1}{m_2}+1})^{i-1}}{(i-1)!} e^{-\lambda t(\frac{m}{k})^{\frac{m_1}{m_2}+1}} \tag{11}$$

Using Equation (11), we can express the average hyperdegree distribution of the clustered hypernetwork as

$$P(k) \approx \lim_{n \to \infty} m_1 E[N(t)] \sum_{ij} P(k_{ij}^h(t) = k) = \frac{1}{m}(\frac{m_1}{m_2} + 1)(\frac{m}{k})^{\frac{m_1}{m_2}+2} \tag{12}$$

We then proceed by assuming that the dynamical mean-field reaction rate equation based on the mean-field theory model [19] can be expressed as

$$\partial_t \rho_k(t) = -\gamma \rho_k(t) + \beta k(m_1 + m_2 - 1)[1 - \rho_k(t)]\Theta(\rho_k(t)) \tag{13}$$

where $\rho_k$ is the relative density of I-state nodes with hyperdegree $k$. The left side of the equation is the rate of change in the density of the node in the I-state with hyperdegree $k$ at time step $t$. The first term on the right is the annihilation term, which indicates that the I-state node with hyperdegree $k$ reverts to the S-state with probability $\gamma$. The second term is the generation term, which indicates that the S-state($1 - \rho_k(t)$) node with hyperdegree $k$ acquires information to become an I-state node with probability $\beta$.

In steady state, $\partial_t \rho_k(t) = 0$, and Equation (13) can be written as

$$\rho_k = \frac{\beta k(m_1 + m_2 - 1)\Theta(\beta, \gamma)}{\gamma + \beta k(m_1 + m_2 - 1)\Theta(\beta, \gamma)} \tag{14}$$

where the probability of a hyperedge linking to an informed node is

$$\Theta(\beta, \gamma) = \sum_k \frac{kP(k)\rho_k}{\sum_s sP(s)} \tag{15}$$

Bringing Equation (14) into Equation (15), we obtain

$$\Theta(\beta, \gamma) = \sum_k \frac{kP(k)}{\sum_s sP(s)} \frac{\beta k(m_1 + m_2 - 1)\Theta(\beta, \gamma)}{\gamma + \beta k(m_1 + m_2 - 1)\Theta(\beta, \gamma)}$$

$$= \int_m^\infty \frac{kP(k)}{\sum_s sP(s)} \frac{\beta k(m_1 + m_2 - 1)\Theta(\beta, \gamma)}{\gamma + \beta k(m_1 + m_2 - 1)\Theta(\beta, \gamma)} dk \tag{16}$$

In Equation (16), $\sum_s sP(s) = \langle k \rangle$, $\langle k \rangle = \int_m^\infty kP(k)dk = \frac{(m_1+m_2)m}{m_1}$. Taking Equation (12) and $\langle k \rangle$ into Equation (16), we obtain

$$\Theta(\beta, \gamma) = \int_m^\infty \frac{m_1}{(m_1 + m_2)m} k \frac{1}{m}(\frac{m_1}{m_2} + 1)(\frac{m}{k})^{\frac{m_1}{m_2}+2} \frac{\beta k(m_1 + m_2 - 1)\Theta(\beta, \gamma)}{\gamma + \beta k(m_1 + m_2 - 1)\Theta(\beta, \gamma)} \tag{17}$$

We simplify Equation (17) to obtain

$$\frac{m_2}{m_1 m^{\frac{m_1}{m_2}} \beta(m_1 + m_2 - 1)} = \int_m^\infty \frac{1}{k^{\frac{m_1}{m_2}}} \frac{1}{\gamma + \beta k(m_1 + m_2 - 1)\Theta(\beta, \gamma)} dk \tag{18}$$

Substituting Equations (12) and (14) into $\rho(t) = \sum_k P(k)\rho_k(t)$, we have

$$
\begin{aligned}
\rho &= \int_m^\infty \frac{1}{m}\left(\frac{m_1}{m_2}+1\right)\left(\frac{m}{k}\right)^{\frac{m_1}{m_2}+2} \frac{\beta k(m_1+m_2-1)\Theta(\beta,\gamma)}{\gamma+\beta k(m_1+m_2-1)\Theta(\beta,\gamma)}dk \\
&= \frac{1}{m}\left(\frac{m_1}{m_2}+1\right)m^{\frac{m_1}{m_2}+2}\beta(m_1+m_2-1)\Theta(\beta,\gamma)\left(\int_m^\infty \frac{1}{\gamma}\frac{1}{k^{\frac{m_1}{m_2}+1}}dk\right. \\
&\left. -\frac{\beta(m_1+m_2-1)\Theta(\beta,\gamma)}{\gamma}\int_m^\infty \frac{1}{k^{\frac{m_1}{m_2}}}\frac{1}{\gamma+\beta k(m_1+m_2-1)\Theta(\beta,\gamma)}dk\right)
\end{aligned}
\tag{19}
$$

From Equation (18), the solution of Equation (19) is

$$
\rho = \frac{m(m_1+m_2)(m_1+m_2-1)\beta\Theta(\beta,\gamma)(1-\Theta(\beta,\gamma))}{\gamma m_1}
\tag{20}
$$

Equation (20) indicates that the informed node density $\rho$ at the steady state is a function that is not influenced by the clustering coefficient $p$ or the time step $t$. Since there is $m_1$ in the denominator of Equation (20), the effect of $m_2$ is greater than the effect of $m_1$. We set the effective propagation rate $\lambda = \beta/\gamma$, and $\lambda$ is proportional to $\rho$. $\rho$ can be calculated from the structural parameters of the hypernetwork as well as the spreading rate and the recovering rate.

## 4. Simulation Results and Analysis

This section is a comparison of theoretical and simulation results, aiming at verifying the hyperdegree distribution of aggregation-phenomenon hypernetworks and exploring the information propagation law.

### 4.1. Hyperdegree Distribution Results

To verify the correctness of the hyperdegree distribution, the parameters were set as follows: $N = 5000$, $m_0 = m_1 = m_2 = 3$, and $p = 0.5$. The simulation plot in double-logarithmic coordinates is shown in Figure 3. The dispersion points represent the simulation results, and the straight lines are the results of the theoretical analysis. The theoretical analysis is consistent with the simulation results and suggests that the hyperdegree distribution follows a linear trend with scale-free characteristics.

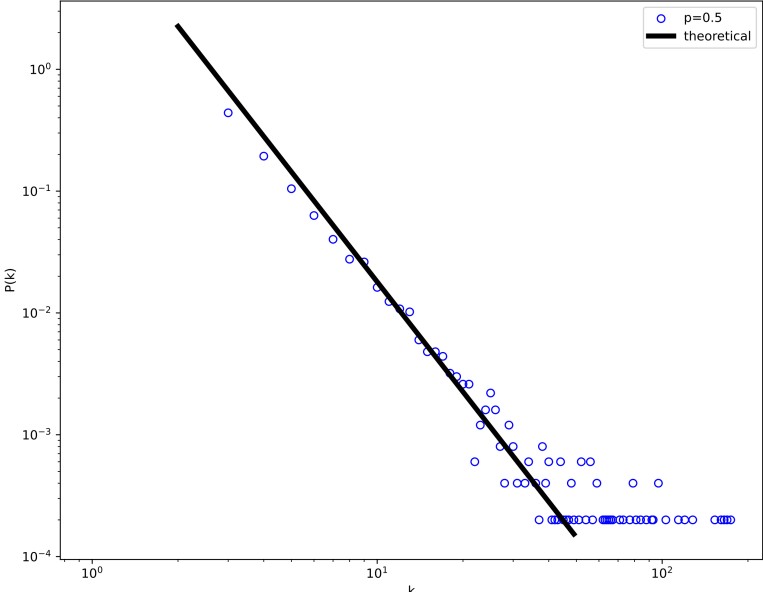

**Figure 3.** Hyperdegree distribution of the hypernetwork with $p = 0.5$. The black line is the result of the theoretical analysis and the blue dots are the simulation results.

Moreover, the hyperdegree distribution shows that the power law index is not influenced by the clustering coefficient $p$. Therefore, we set $p = 0$, 0.2, 0.4, 0.6, 0.8, and 1 for the simulation. The simulation plot in double-logarithmic coordinates is shown in Figure 4. The simulation results of the hyperdegree with different parameters $p$ overlap each other. It can be seen that the power-law exponent of our theoretical prediction of the hyperdegree distribution agrees well with the simulation results. This result shows that the aggregation phenomenon does not affect the scale-free property of the social hypernetworks and supports the argument of Section 3.2 that the hyperdegree distribution is independent of the clustering coefficient $p$.

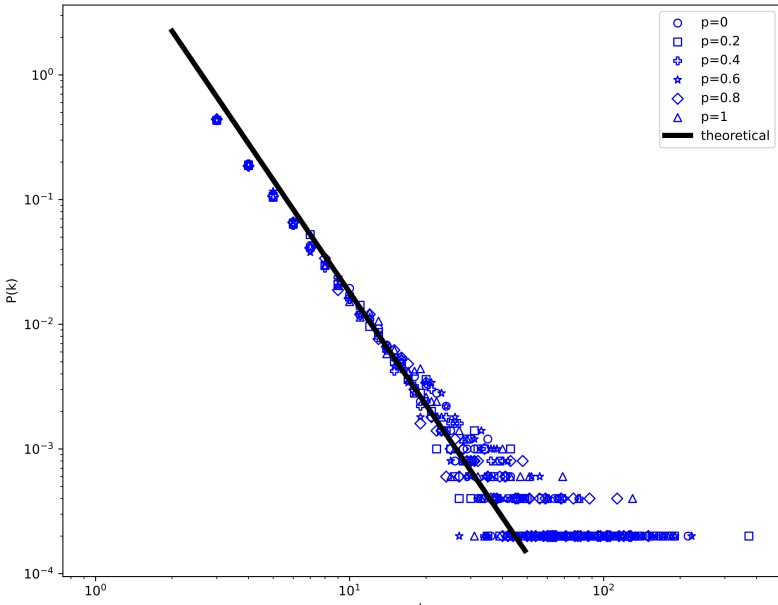

**Figure 4.** The hyperdegree distributions of hypernetworks with varying clustering coefficients $p$. The black line is the result of the theoretical analysis, and the blue dots with different shapes are the simulation results of different clustering coefficients.

### 4.2. The Results of Information Communication Theory Analysis

To verify the validity of the model, the theoretical results were compared with the results obtained from stochastic simulations in the hypernetwork. We also investigated the effects of several parameters, such as the hypernetwork scale, the spreading and recovering rates, the initial propagation seed, and the structural parameters of the hypernetwork, on the information propagation. The hypernetworks were generated based on the construction method described in Section 2.2. Since scholars generally engage in stable collaborations, we assumed that the hypernetwork was static. Initially, one node was selected as the I-state node, and information propagation continued until the system stabilized. To eliminate stochastic effects, the results of each simulation were obtained by averaging the outcomes of 100 independent runs under similar conditions. In order to depict the dynamic process of information flow within the hypernetwork, we determined the trend of the I-state node density as a function of time.

#### 4.2.1. The Simulation Result of the Density of I-State Nodes under Steady State

The values of the parameters were chosen as follows: $m_1 = 1, 3, 5$, $m_2 = 3$, $m = 3$, $\gamma = 0.1$, and $p = 0.6$. The curves in Figure 5 show the theoretical results obtained by using Equation (20), while the density of I-state nodes in the steady state, $\rho$, is depicted by discrete points. Our findings show that information spreads throughout the whole hypernetwork, even if its effective propagation rate $\lambda$ is low. Therefore, the propagation threshold of the RP strategy does not exist.

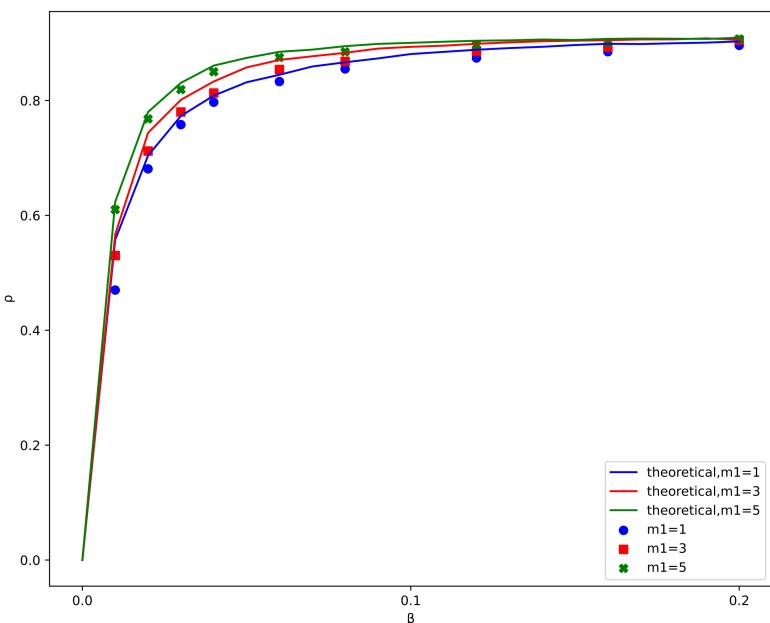

**Figure 5.** Comparison of theoretical analysis and simulation results of informed node density $\rho$ in the steady state.

### 4.2.2. The Impact of Hypernetwork Scale

Figure 6 displays information propagation curves of the density of I-state nodes with respect to time, for various scales of the hypernetwork. The following parameter values were utilized: $m_0 = 6$, $m_1 = 3$, $m_2 = 3$, $m = 3$, $\beta = 0.1$, $\gamma = 0.1$, and $p = 0.6$. The curves demonstrate that the system takes the same time and reaches the same density of I-state nodes in the steady state, regardless of the hypernetwork scale. This result suggests that the hypernetwork scale has a minor impact on information propagation. Consequently, for the following simulation experiments, the number of nodes was set to $N = 1000$.

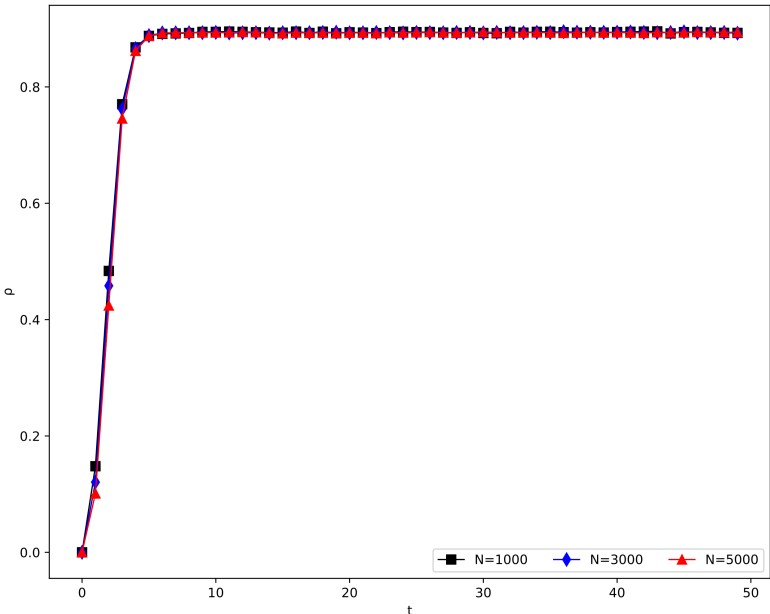

**Figure 6.** Simulation results are displayed for various scales of the hypernetwork.

### 4.2.3. The Influence of the Spreading Seed

Fixed parameters of $m_0 = 6$, $m_1 = 3$, $m_2 = 3$, $m = 3$, $\beta = 0.1$, $\gamma = 0.1$, and $p = 0.6$ were utilized to investigate the effect of the initial spreading seed on the process of propagation. The node with the largest, the average, and the smallest hyperdegree in the hypernetwork were chosen as the initial I-state nodes. The corresponding propagation curves are illustrated in Figure 7. The results show that the spreading seed with the largest hyperdegree is chosen to have the shortest time to reach the steady state. It is noteworthy that the three curves converge to the same value in the steady state. Therefore, the more extensive the socialization of individuals, the faster the information spreads.

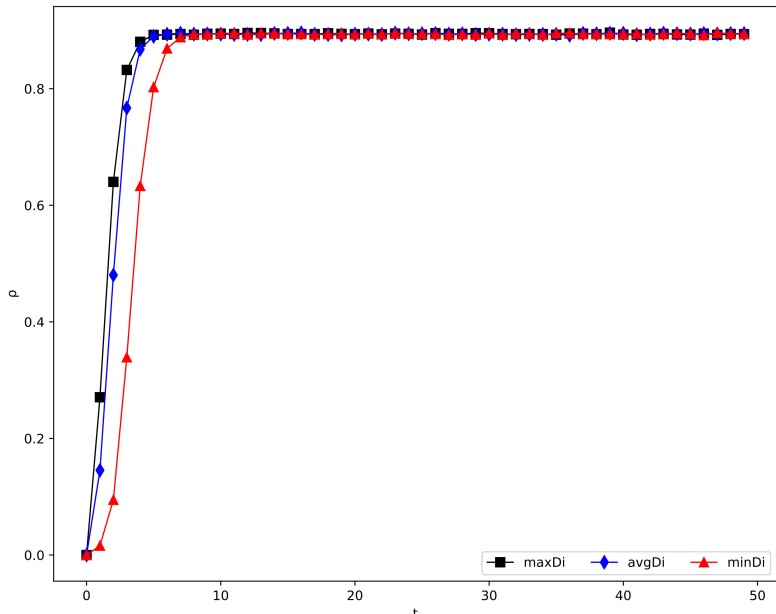

**Figure 7.** Simulation outcomes with various initial spreading seeds.

### 4.2.4. Effect of Clustering Coefficient

To analyze the effect of clustering coefficient $p$ on information propagation, we set the hypernetwork construction parameters as $m_0 = 6$, $m_1 = 3$, $m_2 = 3$, $m = 3$, $\beta = 0.1$, $\gamma = 0.1$, and $p = 0$, 0.2, 0.4, 0.6, 0.8, 1. The information propagation curves for different clustering coefficients are presented in Figure 8. As demonstrated, the information propagation rate slows down with increasing $p$, but all the curves tend to reach the same steady state value. The larger the value of $p$, the fewer direct connections between nodes and therefore, the slower the rate of information dissemination. However, when the steady state is reached, all curves converge to the same value, thus confirming the argument that $\rho$ is independent of $p$ made in Section 3.2. To enhance the scientific validity of the simulation, all subsequent experiments were set with $p = 0.6$ as the clustering coefficient.

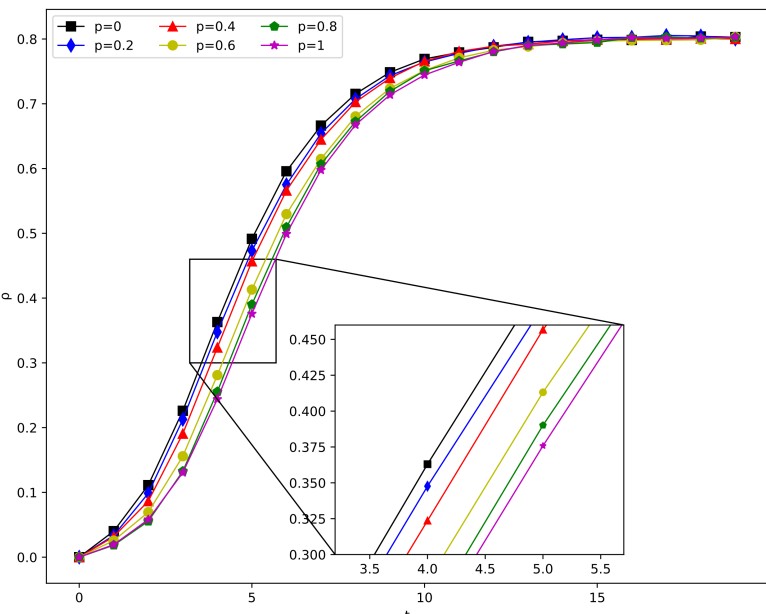

**Figure 8.** Information dissemination curves with different clustering coefficients.

### 4.2.5. Impact of Spreading Rate and Recovering Rate

To assess the impact of spreading rate and recovering rate, the following parameters were set: $m_0 = 6$, $m_1 = 3$, $m_2 = 3$, $m = 3$, and $p = 0.6$.

(i)     Effect of $\beta$

By setting the spreading rate $\beta$ to 0.05, 0.1, and 0.2, while keeping the recovering rate $\gamma = 0.1$ and the other parameters constant, we obtained the information diffusion graph presented in Figure 9. The dissemination rate depicts a person's capability to spread information. The propagation rate affects not only the speed of information dissemination but also the density of I-state nodes at the steady state. The spreading rate is proportional to the propagation speed as well as the steady-state value.

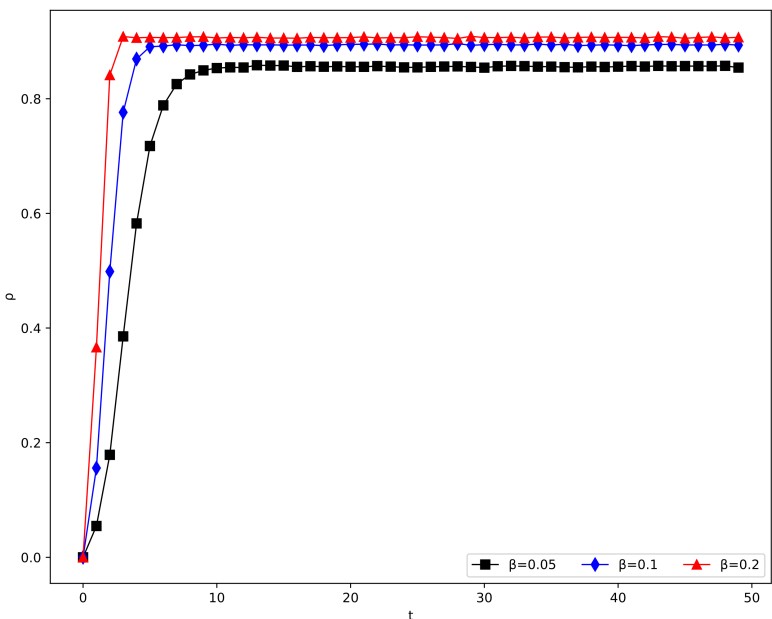

**Figure 9.** Information diffusion curve with various spreading rates.

(ii)    Effect of $\gamma$

We set $\beta = 0.1$ and $\gamma = 0.05, 0.10,$ and $0.20$, to obtain the information propagation curve depicted in Figure 10. The recovering rate reflects a person's ability to resist or forget information. As demonstrated in the figure, the three curves take almost the same amount of time to reach the maximum point, and the slope of the curves hardly varies when the recovering rate changes. This disparity indicates that the recovering rate has a small effect on the rate of information diffusion. Larger recovering rates result in lower densities of I-state nodes at the steady state.

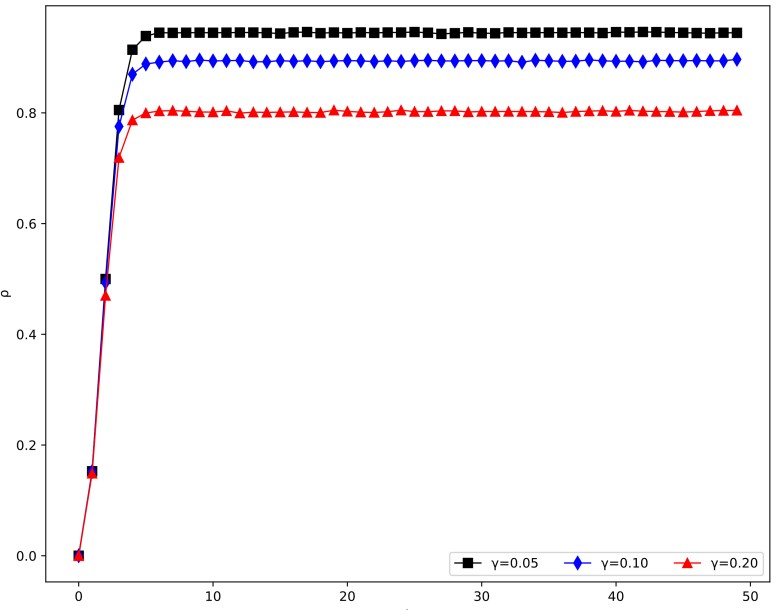

**Figure 10.** Information diffusion curve with various recovering rates.

4.2.6. The Influence of Structural Parameters $m_1$, $m_2$, and $m$

(i)    Effect of $m_1$

In this paper, $m_1$ denotes the number of new nodes that join the hypernetwork at each time step. In order to investigate how $m_1$ influences the propagation process, the following parameters were established: $m_0 = 4$, $m_1 = 1, 3, 5$, $m_2 = 3$, $m = 3$, $\beta = 0.1$, $\gamma = 0.1$, and $p = 0.6$. The information propagation curves for various $m_1$ values are presented in Figure 11. As depicted in the figure, the curves exhibit slight differences, and the information propagation accelerates with increasing $m_1$ values. It can be concluded that an increase in $m_1$ can promote information dissemination, but the change is not very significant.

(ii)    Effect of $m_2$

Here, $m_2$ denotes the quantity of old nodes chosen when new nodes enter the network. To evaluate the effect of $m_2$ on the information spreading process, the following parameters were set: $m_0 = 6$, $m_1 = 3$, $m_2 = 1, 3, 5$, $m = 3$, $\beta = 0.1$, $\gamma = 0.1$, and $p = 0.6$. The information propagation curves for different $m_2$'s are depicted in Figure 12. The increase in the number of selected old nodes means that the average hyperdegree of the supernetwork then increases. As the simulation results indicate, the closer the relationship between individuals and society, the higher the average hyperdegree, the more quickly information spreads, and the more widely information disseminates.

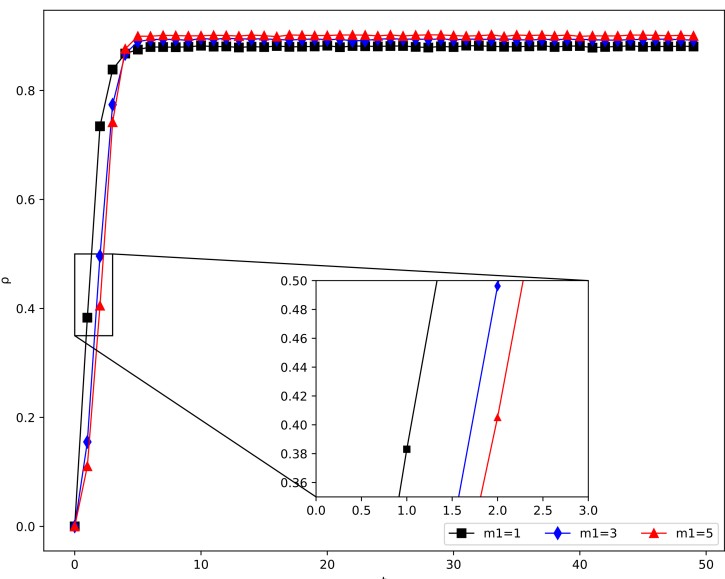

**Figure 11.** Information propagation curves of different $m_1$'s.

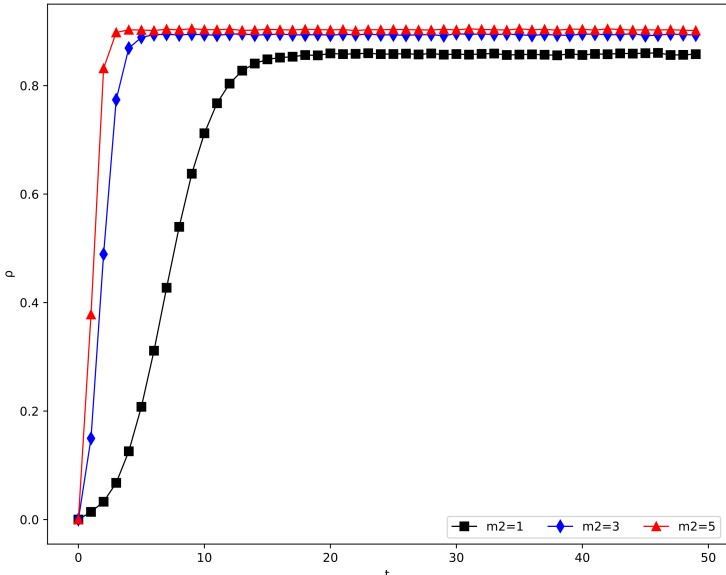

**Figure 12.** Information propagation curves of different $m_2$'s.

(iii)   Effect of $m_1 + m_2$

The number of neighboring nodes for a specific node is directly connected to both the dissemination process and outcomes of information, which are jointly determined by $m_1$ and $m_2$. Therefore, we set the parameters as $m_0 = 6$, $m_1 = 1, 3, 5$, $m_2 = 1, 3, 5$, $m = 3$, $\beta = 0.1$, $\gamma = 0.1$, and $p = 0.6$ to explore the impact of $m_1$ and $m_2$. The information dissemination curves for different $m_1 + m_2$ values are presented in Figure 13a. As demonstrated, the larger the value of $m_1 + m_2$, the shorter the amount of time needed for the information to reach a stable state, and the larger the value of the steady-state phase. Additionally, we set identical values for $m_1 + m_2$ but different values for $m_1$ and $m_2$, as shown below: $m_1 = 1, 3, 5$, $m_2 = 5, 3, 1$. The information propagation curve is presented in Figure 13b. As seen in the figure, the density of I-state nodes is almost the same in the steady-state phase in the three cases. Importantly, for the same $m_1 + m_2$ value, higher $m_2$ values result in a faster system to reach the steady state. This discovery implies that an increased number of

neighboring nodes of I-state nodes accelerates the rate of information propagation. This observation aligns with the theoretical analysis presented in Section 3.2.

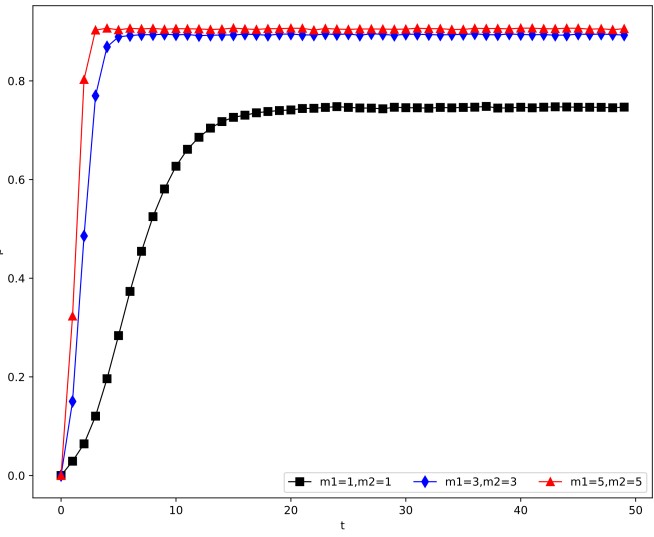

(**a**) $m_1 = m_2$; $m_1 + m_2$ curves have different values.

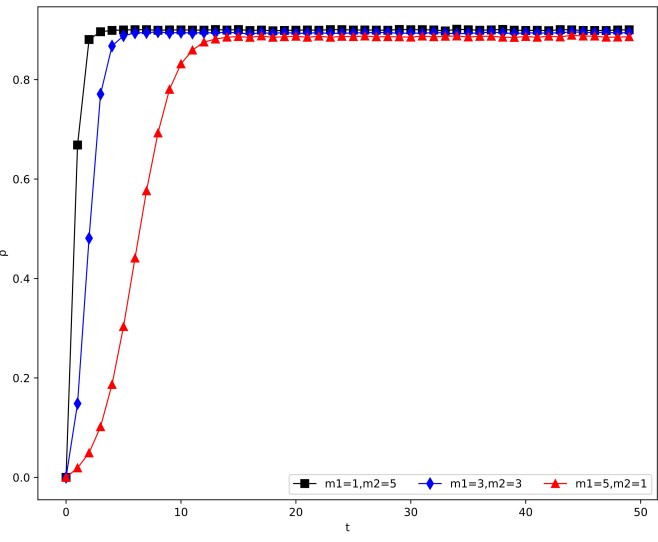

(**b**) $m_1 \neq m_2$; $m_1 + m_2$ curves have the same value.

**Figure 13.** The evolution of information dissemination with different values of $m_1 + m_2$.

(iv)   Effect of $m$

Here, $m$ signifies the number of generated hyperedges generated every time a new node is added. To explore how $m$ affects the propagation process, we set $m_0 = 6$, $m_1 = 3$, $m_2 = 3$, $\beta = 0.1$, $\gamma = 0.1$, $p = 0.6$, and $m = 1, 3, 5$. The information propagation graph is presented in Figure 14. As the number of hyperedges generated per new node addition increases, nodes become more closely connected, and the entire network becomes more interconnected. Therefore, larger $m$ values boost the information propagation efficiency, resulting in higher steady-state values.

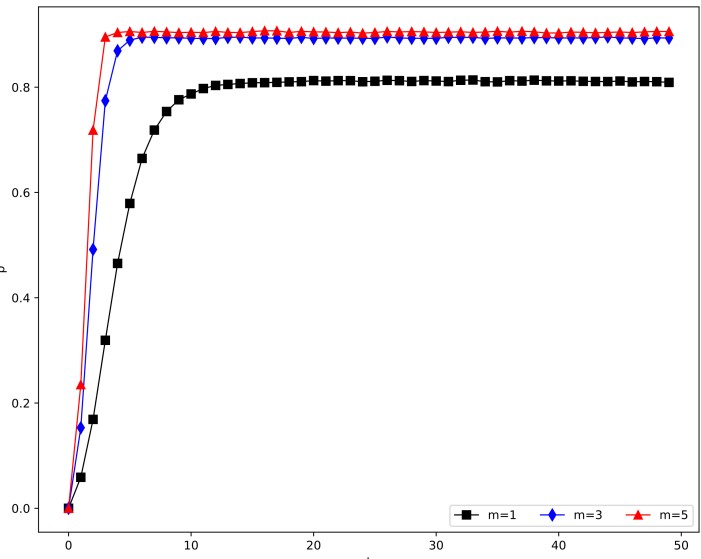

**Figure 14.** Information propagation curves of different *m*'s.

### 4.2.7. Compare with BA Hypernetwork

To analyze the process of information propagation under diverse hypernetwork structures, two hypernetworks were generated with $N = 1000$: the aggregation-phenomenon hypernetwork and the BA scale-free hypernetwork. The network structure parameters were set as $m_0 = 6$, $m_1 = 3$, $m_2 = 3$, $m = 3$, $\beta = 0.1$, and $\gamma = 0.1$, and the clustering coefficient of the aggregation-phenomenon hypernetwork was $p = 0$. Figure 15 illustrates the comparison between the aggregation-phenomenon hypernetwork and the conventional hypernetwork under the same parameters, where the curves of the propagation rate and steady-state values are almost identical. In fact, when the clustering coefficient $p = 0$, the model becomes the BA hypernetwork model, which the simulation results corroborate. Thus, the model introduced in this paper is a generalization of the BA hypernetwork model. Moreover, in social hypernetworks, friends of friends are more likely to build friendships. Consequently, the hypernetwork structure of aggregation phenomena better reflects reality, providing a distinctive advantage that cannot be achieved by BA hypernetworks.

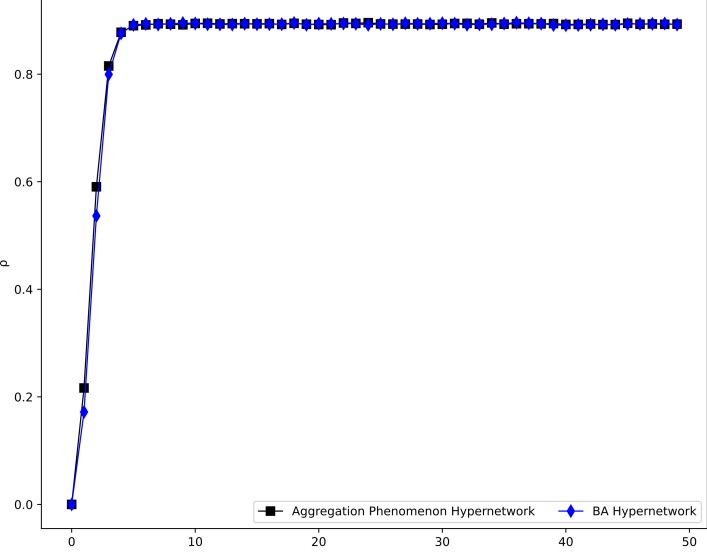

**Figure 15.** Comparison of information propagation curves for BA hypernetwork and aggregation-phenomenon hypernetwork.

### 4.3. Summary of the Experimental Results

To present the experimental results more clearly, we have tabulated the comparison results of various parameters as shown in Table 1.

**Table 1.** Comparison of experimental results for different parameters.

| Parameters | The Density of Informed Nodes ($\rho$) | |
|---|---|---|
| | **Non-Steady State** | **Steady State** |
| Nodes ($N \neq N'$) | $\rho_N \approx \rho_{N'}$ | $\rho_N \approx \rho_{N'}$ |
| Spreading seed | $\rho_{maxDi} > \rho_{avgDi} > \rho_{minDi}$ | $\rho_{maxDi} = \rho_{avgDi} = \rho_{minDi}$ |
| Clustering coefficient ($P < P'$) | $\rho_P > \rho_{P'}$ | $\rho_P = \rho_{P'}$ |
| Clustering coefficient ($P = 0$) | $\rho_{AP}{}^{1} = \rho_{BA}$ | $\rho_{AP} = \rho_{BA}$ |
| Spreading rate ($\beta < \beta'$) | $\rho_\beta < \rho_{\beta'}$ | $\rho_\beta < \rho_{\beta'}$ |
| Recovering rate ($\gamma < \gamma'$) | $\rho_\gamma \approx \rho_{\gamma'}$ | $\rho_\gamma > \rho_{\gamma'}$ |
| New nodes ($m_1 < m_1'$) | $\rho_{m_1} > \rho_{m_1'}$ | $\rho_{m_1} \approx \rho_{m_1'}$ |
| Old nodes ($m_2 < m_2'$) | $\rho_{m_2} < \rho_{m_2'}$ | $\rho_{m_2} < \rho_{m_2'}$ |
| New hyperedges ($m < m'$) | $\rho_m < \rho_{m'}$ | $\rho_m < \rho_{m'}$ |
| $m_1 = m_2, (m_1 + m_2) < (m_1 + m_2)'$ | $\rho_{(m_1+m_2)} < \rho_{(m_1+m_2)'}$ | $\rho_{(m_1+m_2)} < \rho_{(m_1+m_2)'}$ |
| $m_1 \neq m_2, (m_1 + m_2) = (m_1 + m_2)'$ | $\rho_{m_1=*,m_{2min}} < \rho_{m_1=*,m_{2max}}$ | $\rho_{(m_1+m_2)} = \rho_{(m_1+m_2)'}$ |

[1] We abbreviated the aggregation-phenomenon hypernetwork as AP.

The tabulated data reveal that the simulation and theoretical predictions are consistent. Additionally, it is noteworthy that while the clustering coefficient has no effect on the density of informed nodes in the steady state, it does impact the density of informed nodes before the steady state is reached. This means that the clustering coefficient affects the rate of information propagation, with higher clustering coefficients leading to a slower information spread. Importantly, this conclusion cannot be derived from the theoretical analysis alone.

## 5. Conclusions and Prospect

In order to study the characteristics of information propagation in real social hypernetworks, a hypernetwork model and SIS propagation model based on the aggregation phenomenon were constructed. A theoretical analysis and simulation experiments were used to verify the hyperdegree distribution and the information propagation law under the aggregation phenomenon. The theoretical analysis and simulation experiments were in complete agreement. Although the proposed model clustering coefficients had no effect on the scalar law of the hyperdegree distribution, we know that in many real systems, especially social hypernetworks, power-law and high-aggregation phenomena often co-exist. Most notably, the information propagation rate decreased with the increase in the clustering coefficient. The global dissemination of information was not significantly impacted by the magnitude of the hypernetwork scale, while individuals with more neighbors had a higher information dissemination ability, which had a greater impact on the early stage of information dissemination. Both spreading rate and recovering rate affected the information dissemination ability. The larger the spreading rate, the faster the information spread and the larger the steady-state value. The larger the recovering rate, the smaller the steady-state value, and the recovering rate had less influence on the rate of information dissemination. By analyzing the parameters of the network structure, it was found that the larger the number of newly added nodes, the larger the number of old nodes selected, the larger the number of new hyperedges generated each time, and the faster the information propagation. The effect of the number of old nodes was greater than the effect of the number of new nodes.

This paper introduced a new research direction in information dissemination for hypernetwork models and developed a theoretical system for hypernetwork information dissemination, offering innovative ideas for hypernetwork studies. However, further

investigation into the dissemination process is required to complement the current research. Thus, several potential future research directions are proposed as follows:

1. The hypernetwork structure is based on *k*-uniform hypergraphs. The number of individuals in a social hypernetwork is usually different, and it remains to be further investigated whether there are unique information dissemination characteristics on nonuniform hypergraphs based on aggregation phenomena.
2. The information dissemination discussed in this paper is based on a static hypernetwork. However, the cooperative relationships between individuals change according to social activities. Thus, the next consideration is to extend the aggregation-phenomenon hypernetworks to dynamic hypernetworks.
3. The validation of the current theory requires empirical data to support the proposed theoretical model.
4. The information dissemination can be influenced by various intricate factors, including misunderstanding, loss, and addition of information, which will be a key research direction in the future.

**Author Contributions:** Investigation, P.L.; methodology, P.L. and H.D.; resources, F.L. and F.H.; software, P.L. and L.W.; validation, F.L.; formal analysis, P.L.; writing—original draft preparation, P.L.; writing—review and editing, P.L., L.W. and F.L.; supervision, L.W., F.L. and F.H. All authors have read and agreed to the published version of the manuscript.

**Funding:** This work was supported by the National Natural Science Foundation of China (no. 61663041), the Qinghai Science and Technology Planning Project (no. 2023-ZJ-916M), the Tibetan Information Processing and Machine Translation Key Laboratory of Qinghai Province, and the Key Laboratory of Tibetan Information Processing, Ministry of Education.

**Institutional Review Board Statement:** Not applicable.

**Informed Consent Statement:** Not applicable.

**Data Availability Statement:** Not applicable.

**Conflicts of Interest:** The authors declare no conflict of interest.

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
