# Peer review of "Study of Information Dissemination in Hypernetworks with Adjustable Clustering Coefficient"

_applsci, doi:10.3390/app13148212_

Round 1
Reviewer 1 Report
- Motivation and contribution of the work have to be stated in introduction.
- English language has to be clearer and polish.
- Some more literature review has to include in tabular form.
- The results can be represented in the tabular for better understanding.
- The concept of work can be explained better.
- Explain the techniques used in evolution of hypernetwork such as agent-based modeling, network analysis, etc.,
- Give the factors that can affect the evolution of hypernetwork.
- Include the entire derivation of hypernetwork.
9. What is the future scope of this work?
English language has to be clearer and polish
Reviewer 2 Report
Minor changes required in this paper.

Minor changes required in this paper.
Reviewer 3 Report
I read the document, followed the development of the model, reviewed results and conclusions, and found the work interesting.
On the other hand, I cannot establish your contribution according to the results presented in the published article.
https://www.mdpi.com/2076-3417/12/21/10934
I found many similarities in the development and the results. There are only changes in the application and the simulation parameters. However, from my point of view, the behavior of the results is the same.
I suggest explaining the differences with the mentioned article and highlighting them in your proposal. Most of the results are similar to the article already published.
Round 2
Reviewer 3 Report
I think that the legend in Figure 3 is changed.
